# Unravelling Effects of Rosemary (*Rosmarinus officinalis* L.) Extract on Hepatic Fat Accumulation and Plasma Lipid Profile in Rats Fed a High-Fat Western-Style Diet

**DOI:** 10.3390/metabo13090974

**Published:** 2023-08-27

**Authors:** Sidsel Madsen, Steffen Yde Bak, Christian Clement Yde, Henrik Max Jensen, Tine Ahrendt Knudsen, Cecilie Bæch-Laursen, Jens Juul Holst, Christoffer Laustsen, Mette Skou Hedemann

**Affiliations:** 1Department of Animal and Veterinary Sciences, Aarhus University, Blichers Allé 20, DK-8830 Tjele, Denmark; 2IFF—Nutrition Biosciences Aps, Edwin Rahrs Vej 38, DK-8220 Brabrand, Denmark; steffen.yde.bak@iff.com (S.Y.B.); christian-clement.yde@iff.com (C.C.Y.); henrik.max.jensen@iff.com (H.M.J.); tine.ahrendt.knudsen@iff.com (T.A.K.); 3Department of Biomedical Sciences and Novo Nordisk Foundation, Center for Basic Metabolic Research, Faculty of Health and Medical Sciences, University of Copenhagen, Blegdamsvej 3, DK-2200 Copenhagen, Denmark; cecilie.b@sund.ku.dk (C.B.-L.); jjholst@sund.ku.dk (J.J.H.); 4The MR Research Centre, Department of Clinical Medicine, Aarhus University, Palle Juul-Jensens Boulevard 99, DK-8200 Aarhus, Denmark; cl@clin.au.dk

**Keywords:** obesity, metabolomics, proteomics, polyphenol, *Rosmarinus officinalis*

## Abstract

The objective of the study was to investigate the preventive effect on obesity-related conditions of rosemary (*Rosmarinus officinalis* L.) extract (RE) in young, healthy rats fed a high-fat Western-style diet to complement the existing knowledge gap concerning the anti-obesity effects of RE in vivo. Sprague Dawley rats (71.3 ± 0.46 g) were fed a high-fat Western-style diet (WD) or WD containing either 1 g/kg feed or 4 g/kg feed RE for six weeks. A group fed standard chow served as a negative control. The treatments did not affect body weight; however, the liver fat percentage was reduced in rats fed RE, and NMR analyses of liver tissue indicated that total cholesterol and triglycerides in the liver were reduced. In plasma, HDL cholesterol was increased while triglycerides were decreased. Rats fed high RE had significantly increased fasting plasma concentrations of Glucagon-like peptide-1 (GLP-1). Proteomics analyses of liver tissue showed that RE increased enzymes involved in fatty acid oxidation, possibly associated with the higher fasting GLP-1 levels, which may explain the improvement of the overall lipid profile and hepatic fat accumulation. Furthermore, high levels of succinic acid in the cecal content of RE-treated animals suggested a modulation of the microbiota composition. In conclusion, our results suggest that RE may alleviate the effects of consuming a high-fat diet through increased GLP-1 secretion and changes in microbiota composition.

## 1. Introduction

Worldwide, the prevalence of obesity has increased markedly over the last decades, and obesity is now defined as a pandemic. Obesity is linked to a sedentary lifestyle combined with a high intake of energy-dense food products and is associated with an increased risk of numerous comorbidities including type 2 diabetes, certain cancers, cardiovascular diseases, asthma, gallbladder disease, osteoarthritis, and chronic back pain, resulting in higher overall mortality [1].

In recent years, the interest in natural, bioactive compounds with anti-obesity effects has increased. Polyphenols are a group of compounds found in many fruits, vegetables, and cereals [2]. Polyphenols have been studied in relation to human health, and regular consumption is thought to be associated with a decreased risk of several chronic diseases [3,4]. Obesity is one of the conditions, where polyphenols appear to have a preventive effect. Different polyphenols have shown the ability to modulate hormonal appetite regulation [5], inhibit the activity of digestive enzymes [6], inhibit fatty acid synthesis [7], or increase fatty acid oxidation [8], all of which are considered expedient in attenuating the development of obesity.

Rosemary (*Rosmarinus officinalis* L.) is an aromatic, evergreen plant containing high concentrations of polyphenols, with carnosic acid and rosmarinic acid being the most abundant bioactive compounds [9]. In medical use, rosemary has been shown to be beneficial in several conditions, including obesity, metabolic syndrome, and type 2 diabetes, and potential mechanisms of action include an ability to limit absorption of lipids [10] and inhibit hepatic fat accumulation and adipocyte hypertrophy [11]. 

Recent in vitro studies showed that rosemary extract (RE) stimulated the secretion of the satiety-inducing hormone Glucagon-like peptide-1 (GLP-1) from immortalized GLP-1 producing cell lines [12]. GLP-1 is a peptide hormone secreted from intestinal L-cells in response to nutrients in the intestinal lumen. Following secretion, GLP-1 is rapidly inactivated by enzymes in plasma and on cell surfaces, resulting in a half-life of intact GLP-1 of only one to two minutes in humans [13]. Upon entering the blood-stream, GLP-1 is circulated to several target organs, where its metabolic functions include stimulation of insulin secretion, induction of satiety, inhibition of hepatic gluconeogenesis, stimulation of glycogen synthesis and glucose oxidation in muscle tissue, and stimulation of free fatty acid (FFA) synthesis and glucose uptake in adipose tissue, although these effects may be secondary to the actions on the endocrine pancreas or the effects on body weight [14,15,16,17].

Dyslipidemia is often seen in obese patients and includes reduced levels of circulating high-density lipoproteins (HDL) and enhanced levels of low-density lipoproteins (LDL) and triglycerides [18]. Obesity is associated with liver diseases, such as non-alcoholic fatty liver disease (NAFLD). NAFLD is characterized by excessive intrahepatic triglyceride levels and may be accompanied by excessive levels of FFAs and cholesterol in the plasma. It is often associated with alterations in whole-body metabolic homeostasis [19]. Consumption of polyphenols has been associated with the prevention and treatment of NAFLD [20], and the mechanisms may involve the effects of microbial gut metabolites as well as stimulation of gut-derived hormones [21].

We hypothesized that RE reduces feed intake in rats and inhibits diet-induced weight gain through increased fasting GLP-1 levels. The study was carried out in young, non-obese rats with the aim to investigate if daily consumption of a high-fat diet with RE over a period of six weeks influenced weight gain. We studied metabolic parameters related to obesity and used untargeted metabolomics and proteomics to decipher the mode of action.

## 2. Materials and Methods

### 2.1. Plant Material and Extract

The RE (IFF-Nutrient Bioscience ApS, Brabrand, Denmark) used in the present experiment was a dried powder from a water/ethanol extract of rosemary leaves, containing min. 65% carnosic acid and min. 70% carnosic acid + carnosol.

### 2.2. Animals

In vivo study: The study included 64 male Sprague Dawley rats obtained from Taconic Biosciences (Ejby, Denmark) with a starting weight of 71.34 g (±0.46 SEM). The rats were divided into pairs of two rats (32 pairs) and the pairs were divided into four experimental groups (8 pairs per group). Each group was fed one of four pelleted experimental diets; standard rodent chow (Altromin 1324, Altromin GmbH, Lage, Germany) (ST), Western-style diet (D12079B; 40% fat, 43% carbohydrates, 17% protein; Research Diets, New Brunswick, NJ, USA) (WD), WD supplemented with 100 mg rosemary extract (RE) pr. 100 g feed (WD+low-RE), or WD supplemented with 400 mg RE pr. 100 g feed (WD+high-RE). The dose of RE added was based on a previous experiment where reduced weight gain was observed in mice fed a high-fat diet supplemented with RE [22]. The rats were expected to eat 10–20 g feed/day which corresponds to daily intakes of 40–80 mg of RE in rats fed the highest level of RE. The chemical composition of the diets was determined as previously described [23] and is shown in Appendix A.

Animals were housed in pairs in clear plastic boxes (38 × 60 cm) bedded with shavings and followed a 12:12 h light-dark cycle, at 25 °C and 60% humidity, with ad libitum access to water and feed during the entire experiment. Rats were individually marked to enable measurement of individual body weight (BW). BW gain and feed intake were measured twice weekly (see Appendix A). The experiment was conducted over 46 days. 

### 2.3. Sample Collection

At the end of the experiment, and after six hours of food deprivation to allow for the normalization of metabolic markers, all rats were euthanized by an intraperitoneal injection of 0.5 mL pentobarbital (200 mg/mL) with lidocaine (20 mg/mL). From each box, one rat was euthanized and kept intact at −20 °C for whole body composition analysis, while the second rat in the box was used for tissue and blood collection. When death was ensured, blood was collected from the heart into EDTA-coated tubes containing 1% (*V*/*V*) dipeptidylpeptidase-4 (DPP-4) inhibitor (Merck KGaA, Darmstad, Germany). Blood samples were immediately centrifuged at 2000× *g* for 12 min at 4 °C before plasma was stored at −80 °C until further analysis. The abdominal cavity was opened, and the liver was dissected out, weighed, and snap-frozen in liquid nitrogen. Samples were collected from three segments of the small intestine (5 cm distal to the pylorus sphincter, a mid-section, and 5 cm proximal to the ileocaecal valve). The samples were washed thoroughly in saline before being placed in RNAlater® (ThermoFisher Scientific, Waltham, MA, USA), and further treated according to the manufacturer’s protocol. All tissue samples were stored at −80 °C until further analysis. The caecum was dissected free and cecal tissue and content were weighed. Content was collected and stored at −80 °C until further analysis.

### 2.4. Studies on Isolated Perfused Rat Small Intestine 

Male Wistar rats (~250 g) were obtained from Janvier (Saint Berthevin Cedex, France) and housed with ad libitum access to standard chow and water, following a 12:12 h light:dark cycle. Rats were acclimatized for a least a week before the experiments. On the day of the experiment, rats were transferred to our perfusion facility and anesthetized, and the entire small intestine was isolated and perfused as described in detail in the Appendix A and elsewhere [24,25].

### 2.5. Total Body Fat Content

Body composition data were acquired using a 9.4T pre-clinical MR scanner (Agilent, Yarnton, UK) equipped with a 72 mm proton volume coil (RAPID Biomedical GmbH, Rimpar, Germany). Full body proton NMR spectra from each animal were Fourier Transformed and spectrally analyzed by fitting the water and the fat peaks using a general linear model approach that was developed in-house (MATLAB, The MathWorks, Inc., Natick, MA, USA). Subsequently, the fitted peak areas were used to calculate the fat percentage by division by the water component and fat component signal areas.

### 2.6. Fasting Total GLP-1 Plasma Concentrations

Fasting total GLP-1 concentrations were measured using an in-house radioimmunoassay, as previously described [26], based on an antiserum directed against the amidated C-terminus of GLP-1. The antibody reacts equally with intact amidated GLP-1 (7–36 amide), the primary metabolite (9–36 amide), and truncated sequences produced by mid-site cleavage (e.g., by NEP 24.11). Therefore, the concentrations represent the total GLP-1 concentration. I125-labeled GLP-1 7-36NH2 (a kind gift from Novo Nordisk A/S) was used as a tracer. The experimental detection limit was 1 pmol/L, and the coefficient of variation was <6% at 20 pM.

### 2.7. RNA Extraction and Gene Expression 

Total RNA was extracted from the whole wall of the small intestine using a NucleoSpin RNA kit (Macherey-Nagel GmbH & Co., Ltd., KG, Duren, Germany), in accordance with the manufacturer’s protocol. RNA concentration and purity were evaluated using a NanoDrop spectrophotometer (ND-1000, NanoDrop Technologies, Wilmington, DE, USA). A High-Capacity cDNA Reverse Transcriptase Kit (Life Technologies, Taastrup, Denmark) was used for cDNA synthesis. The qPCR of glyceraldehyde-3-phosphate dehydrogenase (GAPDH), β-actin, and proglucagon (GCG) genes was run in 384-well plates with a 1:4 ratio of cDNA and TaqMan Master Mix (Applied Biosystems, Foster, CA, USA), according to the manufacturer’s protocol. The analyses were performed in triplicates for the housekeeping genes and in duplicates for the target genes, under standard amplification conditions determined for the Applied Biosystem ViiA 7 Real-Time PCR system (Life Technologies, Taastrup, Denmark). Expression levels were first normalized to the expression level in the most proximal sample in the respective rats, and values were then normalized to the control group fed the standard diet. For the interaction between the treatment and the tissue, normalization to the most proximal segment for the standard diet treatment was used.

### 2.8. Plasma Metabolic Markers

Plasma cholesterol, HDL, LDL, and triglycerides were quantified by standard procedures (Siemens Diagnostics^®^ Clinical Methods for ADVIA 1800). FFAs were determined using the Wako, NEFA C ACS-ACOD assay method. All analyses were performed using an autoanalyzer, ADVIA 1800 Chemistry System (Siemens Medical Solutions, Tarrytown, NY 10591, USA).

### 2.9. Analysis of Hepatic Fat Percentage 

Liver samples were homogenized in twice the amount of methanol (*w*/*v*) using an Ultra-Turrax homogenizer (IKA Labortechnik, Staufen, Germany) while placed in an ice bath. Of this homogenate, 600 mg was mixed with 1.0 mL water, 1.5 mL methanol, and 1.0 mL chloroform, before being shaken for one minute. Water (1.0 mL) and chloroform (2.0 mL) were added to the mixture, following shaking for one minute. Phase separation was accelerated by centrifugation at 2000× *g* for 5 min. The lower chloroform phase (2 mL) was collected and dried, and the residue was weighed for determination of fat content.

### 2.10. Proteomics Analysis of Liver Tissue

Approximately 100–200 mg of tissue was excised from the left lateral lobe. Frozen tissue was covered with 1 mL lysis buffer (5% SDS in 100 mM triethylammonium bicarbonate, TEAB, pH 7.5). Tissue homogenization was done using a ball mill at 25 Hz for 10 min (Retsch GmbH, Haan, Germany). Homogenates were centrifuged for 10 min at 14,000× *g*. The supernatants were heated to 95 °C for 10 min to fully denature proteins. DNA was sheared using a Bioruptor Pico (Diagenode Diagnostics, Liege, Belgium) high-energy water bath sonicator system using 10 cycles of 30 s sonication followed by a 30 s break. The temperature was set to 5 °C throughout the entire process. Protein concentrations were estimated in triplicates using a BCA micro assay kit (Thermo Scientific Pierce, Rockford, IL, USA). 

Liver lysates were digested into peptides using the S-trap™ micro digestion protocol [27] using a trypsin-to-protein ratio of 1:30. Peptide elutes were dried down and resuspended in 50 µL 0.1% TFA. Concentrations of the peptides were estimated using a micro-BCA™ protein assay kit (Thermo Scientific Pierce, Rockford, IL, USA) and samples were adjusted to equal concentrations. A QC sample (mixture of all samples) was prepared by taking 10 µL from all samples and mixing well. Data acquisition and analysis were performed as described in the Appendix A.

Data Acquisition: Nano LC−MS/MS analyses were performed using an UltiMate™ 3000 RSLCnano System (Thermo Fisher Scientific, Waltham, MA, USA) interfaced to a Q Exactive™ HF Hybrid Quadrupole-Orbitrap™ Mass Spectrometer (Thermo Fisher Scientific, Waltham, MA, USA). Samples were dissolved in 0.1% TFA and loaded onto a 20 mm nanoViper Trap Column (Acclaim™ PepMap™ 100 C18, 3µm particle size, and an i.d. 0.075mm) connected to a 400 mm analytical column (PepSep, ReproSil 3 µm C18 beads, pore diameter 120Å and an i.d. 0.075 mm). Separation was performed at a flow rate of 300 nL/min using a 40 min gradient of 0−41% Solvent B (100% ACN, 0.1% FA) into the Nanospray Flex Ion Source (Thermo Scientific). The Q Exactive HF instrument was operated in a data-dependent MS/MS using HCD fragmentation. The peptide masses were measured by the Orbitrap (MS scans were obtained with a resolution of 60,000 (FWHM) at *m*/*z* 200). The top 12 most intense ions were selected and subjected to fragmentation. Ions were isolated by the quadrupole using a 1.2 Da width isolation window. Fragment spectra were recorded in the Orbitrap with a resolution of 30,000 (FWHM) at *m*/*z* 200. Dynamic exclusion was enabled with an exclusion duration of 30 s, and an exclusion mass tolerance width of ±10 ppm relative to masses on the list. Samples were analyzed in a randomized order with a QC sample after every fifth sample and with five QC samples at the beginning of the sample sequence. 

Data analysis: The LC−MS/MS data were processed (smoothing, background subtraction, and centroiding) using Proteome Discoverer (Version 2.2, Thermo Scientific). The processed LC−MS/MS data were subjected to database searching against a UniProt Rattus Norvegicus database containing 29,944 sequences using an in-house Mascot server. Trypsin was chosen as an enzyme with a maximum of two missed cleavages allowed. S-Carbamidomethyl cysteine was defined as a fixed modification and oxidation of methionine as a variable modification. The MS/MS results were searched with a peptide ion mass tolerance of ±10 ppm and a fragment ion mass tolerance of ±0.8 Da. A percolator [28] was used for calculating the false discovery rate. Only peptides that were identified as rank 1 peptides and with a confidence value of 1% (q < 0.01) were considered for further analysis. 

All raw files were imported into Expressionist v.12.0.9 (Genedata, Basel, Schwitzerland) for data analysis. Imported files were noise filtered using a chemical noise subtraction. Data were RT aligned using a pairwise alignment, filtered, and smoothed before peak detection, based on volumes. Detected peaks were isotopically clustered and singletons were filtered out. Peak clusters were matched with Proteome Discoverer identifications and peptides were grouped based on protein identifications. Proteins were quantified based on the three most intense peptides’ Hi-3 [29]. Quantitative results were exported into Analyst v.12.0.9 (Genedata) for normalization, statistical filtering, and testing. Principle Component Analysis was performed using the GeneData Analyst module. A total of 1570 proteins were identified and quantified and within these, 999 proteins were quantified with a minimum of two peptides per protein. The average finding percentage of the 999 proteins throughout all 44 runs was 99.3%, giving an almost complete data matrix. Data were normalized using an intensity drift normalization, where intensities from QC samples are used to correct for drift in the LC-MS/MS performance. Using an ANOVA, we found 247 proteins significantly changing between one of the four diets with an overall FDR of 5%. Regulated proteins were manually annotated. 

### 2.11. NMR Spectroscopy

A dual-phase methanol/chloroform/water (1:1:1) extraction protocol was performed on the left lateral lobe liver samples (weight 259 ± 94 mg), as previously described [30]. The NMR measurements were performed on a 600 MHz Bruker Avance III spectrometer operating at a frequency of 600.13 MHz for 1H nucleus (Bruker Biospins, Rheinstetten, Germany) and equipped with a 5mm TCI CryoProbe. Polar extracts: 1H NMR spectra were recorded at 298 K using a 1D Noesy pulse sequence with pre-saturation. A total of 256 scans collected into 64 K data points were acquired with a spectral width of 20.02 ppm, a recycle delay of 5 s, and an acquisition time of 2.73 s. Non-polar extracts: 1H NMR spectra were obtained at 298 K using a single pulse sequence with a 30° flip angle. A total of 128 scans collected into 64K data points were acquired with a spectral width of 20.02 ppm, a recycle delay of 3 s, and an acquisition time of 2.73 s. The anomeric glycogen H-1 signal at 5.4 ppm in the polar extracts was manually quantified and normalized to the weight of the liver for each rat. Hepatic lipid signals in the non-polar phase were mainly tentatively assigned according to Fathi et al. [31]. Prior to data analysis, the residual water signal was excluded from the “Polar” spectra, and all 1H NMR spectra were normalized to the weight of each sample. Baseline correction by distribution-based classification [32], alignment using icoshift version 1.3.1 [33], and binning according to an optimized bucketing algorithm [34] were performed for both data sets. Multivariate data analysis was applied using Principal Component Analysis (PCA) on Pareto-scaled data using PLS Toolbox 8.7 (eigenvector Research, USA) in Matlab R2018b (The MathWorks, Inc., Natick, MA, USA) to detect clustering behavior.

### 2.12. Metabolomic Analysis of Plasma and Cecal Contents

LC-MS was used to analyze the plasma and cecal metabolome in an untargeted approach. Plasma samples (150 µL) were placed in the wells of a 96-well plate and deproteinized with 450 µL ACN containing internal standards (*p*-chlorophenylalanine and glycocholic acid (Glycine-1 13C), final concentration 0.01 mg/mL) and processed as previously described [35]. Cecal contents were freeze-dried and milled prior to analysis. Cecal content (50 mg) was mixed with 500 µL 80% ACN with added internal standards as described above. The samples were vortexed for 20 min at room temperature and centrifuged for 10 min at 10,000× *g* and 4 °C. The supernatant was transferred to a 96-square well, 0.45 µM polypropylene filter plate and aspirated into a new plate. The filtered supernatant was treated as described for the plasma samples. UPLC-MS analysis was performed using a Dionex UltiMate 3000 (Dionex, Sunnyvale, CA, USA) ultra-high pressure liquid chromatography system coupled with an Impact HD Quadrupole Time-of-Flight (QTOF) mass spectrometer (Bruker Daltonics GmbH, Bremen, Germany) operating in positive electrospray ionization mode (ESI+) and negative electrospray ionization mode (ESI-) using the instrumental parameters previously described [35,36]. The chromatographic behavior of the samples, pre-processing, and identification of metabolites was assessed as previously described [35].

### 2.13. SCFA Analysis 

Short-chain fatty acid (SCFA) concentrations were measured as previously described [37], with some modifications as described in the Appendix A.

### 2.14. Statistical Analyses

Differences in fasting plasma GLP-1 concentrations, metabolic markers in plasma, total and hepatic fat accumulation, weight, and SCFA were assessed by One-way ANOVA with Tukey multiple-comparison correction. Differences in the presence of liver proteins were assessed by One-way ANOVA using Storey-Tibshriani correction. *p* < 0.05 was considered statistically significant whereas a *p*-value between 0.05 and 0.10 was considered as an indication of a trend towards significance.

Gene expression data were analyzed using a normal mixed model:Yij = μ + αi + υj + εij,
where Yij is the gene expression; μ is the overall mean; αi is the effect of treatment (i = ST, WD, WD+low-RE, WD+high-RE); υj is the random effect of a block (j = 1, 2); and εij is the residual error component. The data are presented as least square means with a 95% confidence interval. Effects were considered significant when *p* < 0.05, whereas 0.05 ≤ *p* ≤ 0.10 represented a tendency.

For perfusion studies, GLP-1 total outputs were calculated by multiplying perfusion flow (7.5 mL/min) with the GLP-1 concentration in the perfusion effluents (measured in pmol/L). Hormone outputs (fmol/min) are presented as means ± SEM. To test for the statistical significance of responses, total GLP-1 outputs were calculated by summing up the outputs during the entire period of stimulus administrations (15 min intra-luminal/intra-vascular RE administration) and were compared to respective total GLP-1 outputs at baseline conditions. Baseline outputs were calculated over a similar number of samples with half of the samples taken immediately before stimulus administration and the other half coming from the terminal part of the subsequent baseline period. Statistical calculations were performed using GraphPad Prism 7 software (La Jolla, CA, USA), employing One-way ANOVA for repeated measurements followed by the Bonferroni multiple comparison test or student *t*-test as indicated in the figure legends. *p* < 0.05 was considered significant. Prior to testing, the D’Agostino-Pearson omnibus normality test was performed to confirm the Gaussian distribution of the data. 

## 3. Results

### 3.1. Body Weight and Feed Intake

As presented in Figure 1A, during the six-week experimental period, a clear effect of time (*p* < 0.0001) was observed for body weight, while no overall effect of treatments was observed (*p* = 0.10). The treatments influenced the 24-h feed intake significantly (*p* = 0.006; Figure 1B). When determining the daily weight gain pr. energy intake (g/kcal), an overall effect of treatment was observed (*p* < 0.0001; Figure 1C). When comparing this for the different treatments, a significant difference was seen on days 3–17; however, no overall effect of treatment was observed after day 13.

### 3.2. Liver Weight and Fat Accumulation in the Liver and Whole Body

Mean total body fat percentages (Figure 2A) following six weeks of treatment were 14.1% (ST), 17.3% (WD), 18.9% (WD+low-RE), and 15.0% (WD+high-RE), and no difference between the treatments was observed (*p* = 0.18). Rats treated with WD had significantly higher liver fat percentages compared to ST treatment (*p* < 0.0001). Treatment with WD+high-RE strongly decreased the fat mass in the liver (Figure 2B), with an average reduction of 46% compared to the WD. The mean weights of wet liver tissue (Figure 2C), expressed as a percentage of total body weight, were 3.99% (ST), 4.50% (WD), 5.26% (WD+low-RE), and 5.16% (WD+high-RE). ST treatment resulted in the lowest relative liver weight, it increased when feeding WD, and the addition of RE to the diet increased the relative liver weight further.

### 3.3. Plasma Metabolic Markers

An overall effect of treatment was observed for total cholesterol (*p* = 0.0008; Figure 3A). Thus, feeding WD, WD+low-RE, and WD+high-RE caused a significantly higher level of total plasma cholesterol, compared to ST, but no significant differences were observed between the three. For mean plasma levels of LDL-cholesterol (Figure 3B), only WD+low-RE treatment resulted in a significantly higher level than ST, whereas WD+high-RE was similar to ST and WD was intermediate. Analysis of mean plasma levels of HDL-cholesterol (Figure 3C) showed that WD+low-RE and WD+high-RE treatment caused significantly higher plasma HDL levels than ST treatment. An effect of treatment on the plasma level of triglycerides was observed (*p* = 0.0002; Figure 3D). The mean plasma levels of plasma triglycerides were significantly higher with WD and WD+low-RE compared to the ST, whereas levels were intermediate in rats fed WD+high-RE. Plasma FFA levels (Figure 3E) were significantly higher for WD compared to ST (*p* = 0.005), whereas WD-low-RE and WD+high-RE were intermediate.

### 3.4. Fasting GLP-1 Plasma Concentrations and Proglucagon Expression

An effect of treatment was observed for the fasting plasma GLP-1 concentration (*p* = 0.01, Figure 4A). Thus, fasting GLP-1 plasma concentrations were increased by WD+high-RE treatment compared to ST and WD treatment, but concentrations were not different from the WD+low-RE group and also did not differ from ST and WD. 

An interaction between treatment and segment was observed for relative proglucagon expression (*p* = 0.03; Figure 4B). The interactive effect was observed in the lower SI where the relative expression was higher in rats fed WD+low-RE and WD+high-RE compared to rats fed ST. For segment-specific changes in relative proglucagon gene expression, the mid-part and the distal part of the small intestine were 21-fold and 48-fold higher than the proximal part, respectively.

### 3.5. Acute Effects of RE on GLP-1 Secretion from Isolated Perfused Rat Small Intestine

The baseline GLP-1 output was (mean ± SEM) 171 ± 11.2 fmol/min. GLP-1 output was neither affected by intra-luminal RE infusion (*p* > 0.99 between total output at 15 min baseline and 15 min stimulation), nor by intra-arterial RE infusion (*p* = 0.97 between preceding total baseline output and 15 min of stimulation) (n = 6, Figure 5). Confirming responsiveness of the perfused preparation, intra-arterial taurodeoxycholic acid (100 µM, a conjugated secondary bile acid) instantly and reversibly increased secretion by a factor of 2–3 during 15 min of stimulation (*p* < 0.05 compared to preceding output at baseline).

### 3.6. Proteomic Analysis of Liver Tissue

In this study, we used label-free LC-MS/MS-based proteomics. A principal components analysis (PCA) score plot of the liver proteins is shown in Figure 6. Principal component (PC) 2 shows the second largest variation (10.5%) in the dataset and, in this case, separates the different diets.

A number of proteins involved in lipid metabolism and oxidation of lipids were found to be affected in WD-fed animals compared to ST-fed animals. The proteins found related to fatty acid metabolism and oxidation and regulated in comparison to the ST diet are shown in Table 1. Interestingly, a number of these proteins are regulated differentially in response to the RE supplement. The hydroxyacyl-coenzyme A dehydrogenase and all-trans-retinol 13,14-reductase (RETSAT) were upregulated by WD whereas, in rats supplemented with RE, these proteins were not significantly altered compared to ST.

The mitochondrial protein Enoyl-[acyl-carrier-protein] reductase and the protein macrophage migration inhibitory factor (MIF) increased in the WD+high-RE diet compared to ST, whereas WD only enhanced the abundance slightly. Changes were also observed in the mitochondrial protein MICOS complex subunit Mic60 which increased by 1.7-fold in the WD+high-RE diet compared to ST. Stearoyl-CoA desaturase 1 (SCD1) was found to be the most regulated protein in response to WD as the protein was 9.8-fold upregulated in the WD samples compared to ST samples. In WD+high-RE treatment, this increase was slightly lower at 7.6-fold.

Several members of UDP-glucuronosyltransferases and glutathione reductases and glutathione S-transferases belonging to the Xenobiotic pathway phase II response were also found to be regulated in response to a high-RE diet (Appendix A). 

### 3.7. NMR Analysis of Liver Tissue

A PCA model on the non-polar liver extracts resulted in a clear separation of the treatments along PC1 (Figure 7A). The WD+low-RE treatment could not be separated from WD in PC1, whereas WD+high-RE showed a shift towards the lipid profile of the ST-fed rats. The PC1 loadings for the hepatic free and total cholesterol could be ranked according to treatments: WD and WD+low-RE > WD+high-RE > ST (Figure 7B). The same pattern was also shown for glycerol backbone signals, which could indicate less hepatic triglyceride in WD+high-RE rats compared to WD rats. The PC1 loadings in Figure 7B of the fatty acyl chain signals have in general both positive and negative loadings for the same functional group (e.g., (CH2)2 at 1.24–1.37 ppm). This could be seen as minor differences in the lipid signals when comparing different treatments. Moreover, the unsaturated fatty acid broad peak from 5.23 to 5.38 ppm shows mainly negatively valued loadings, which means that livers from WD+high-RE rats have a higher prevalence of unsaturated fatty acids than livers from WD and WD+low-RE rats.

The hepatic glycogen levels were quantified in the polar liver extracts by NMR spectroscopy (Figure 2D). A significant treatment effect was found for glycogen hepatic levels with elevated levels in ST and WD+high-RE compared to WD and WD+low-RE rats.

### 3.8. LC-MS Metabolomics of Plasma and Caecum Content

The PCA scores plot of cecal digesta samples analyzed in negative mode revealed four separated clusters based on treatment (Figure 8A). Likewise, the PCA score plot of plasma (Figure 8B) in negative mode showed a clear separation between rats fed ST and WD. The separation between WD and WD with added RE was less clear; however, a ranking of the samples was evident.

In cecal content, the ST-treated rats differed from WD-, WD+low-RE-, and WD+high-RE-treated rats by having high levels of metabolites linked to whole grain intake, e.g., enterolactone and dihydroferulic acid (Appendix A, Appendix A). Moreover, ST-treated rats had higher levels of certain lipid metabolites, indicating different lipid sources in ST and WD. In general, the treatment with WD, WD+low-RE, and WD+high-RE caused high levels of putatively identified bile acid (BA) metabolites (Appendix A). However, for many of these BA metabolites, the concentration was decreasing with the increase in RE concentration. Cholic acid, a primary BA, appears to be the only identified BA metabolite increasing with RE concentration. Many metabolites were seen increasing with RE concentration, most of which were connected to rosemary. These include carnosol, carnosic acid, and glucuronides of these.

In plasma, many RE-derived metabolites, including carnosol, carnosic acid, and rosmadial, were found in WD+low-RE- and WD+high-RE-treated rats, with a clear positive correlation to concentration in the diets (Appendix A, Appendix A). Furthermore, lysophosphatidylcholines comprised a major part of the metabolites differing in the plasma metabolome of rats fed the experimental diets.

### 3.9. Short-Chain Fatty Acids

ST-fed rats had significantly (*p* < 0.0001) more content in the caecum than rats fed WD, who on the other hand had less cecal content than rats fed WD supplemented with RE (Table 2). The total amount of carboxylic acids in the caecum was significantly higher (*p* < 0.0001) in ST-treated rats compared to treatment with WD, WD+low-RE, and WD+high-RE. The same trend was seen in the pool size of acetic, propionic, and butyric acid. In contrast, the pool size of succinic acid was significantly lower in ST- and WD-treated rats compared to treatment with WD+low-RE and WD+high-RE. 

## 4. Discussion

Over the six-week intervention, we did not observe an effect on body weight after supplementing a Western-style diet high in fat with RE, despite an elevated fasting total GLP-1 concentration, and hence our hypothesis was rejected. In a previous study [10], daily administration of carnosic acid (20 mg/kg BW/day), a polyphenol found in high concentrations in rosemary, to six-week-old mice fed a high-fat diet ad libitum for 14 days was sufficient to reduce weight gain significantly. For comparison, the WD+high-RE rats consumed up to approximately 150 mg RE per kg body weight per day, corresponding to 98 mg of carnosic acid per kg body weight per day. Thus, the lack of effect of the treatment on body weight in the present experiment is not likely to be due to insufficient dosing. Furthermore, adult, obese mice on a high-fat diet supplemented with 200 mg/kg body weight of RE had a significantly lower weight gain after 35 days of treatment compared to those who did not receive RE [38]. The lack of effect on BW observed in the present study may be explained by the choice of animal and strain. In a study with rats with a start weight of approximately 100 g, feeding the rats either a standard chow or a high-fat diet did not induce any diet-related differences in body weight over a period of six weeks in Sprague Dawley rats, while in Long-Evans rats, the animals exposed to a high-fat diet gained significantly more weight [39]. In addition, for outbred Sprague Dawley rats fed a diet high in fat and energy, only about half will develop obesity, while the other half is resistant to diet-induced obesity and will gain weight at the same pace as standard chow-fed controls [40].

We showed that feeding WD+high-RE for six weeks significantly increased fasting total GLP-1 plasma concentrations. This is not supported by our studies on isolated perfused rat small intestine, where intra-luminal or intra-arterial RE infusion did not stimulate the secretion of intestinal GLP-1. However, our previous in vitro studies showed augmented secretion of GLP-1 from STC-1 and HuTu-80 cells when stimulated with RE [12]. The discrepancy between the long-term and acute effects of RE on GLP-1 secretion points towards an adaptive effect of RE, as previously seen for nutritional modulation of GLP-1 secretion [41]. It is well documented that RE is a potent inhibitor of the primary enzyme catalyzing GLP-1 degradation, DPP-4 [42]. Our metabolomics data demonstrated that several RE-derived metabolites, including carnosol, carnosic acid, and rosmadial, were present in plasma which could possibly lead to inhibition of DPP-4 and thereby a higher concentration of active GLP-1. Confirmation of this would require measurement of active GLP-1. We observed a higher proglucagon gene expression after treatment with WD+low-RE and WD+high-RE, but this does not correlate with the increased fasting levels after WD+high-RE only. The expression increased progressively from the proximal duodenum to the distal ileum, which corresponds to the tissue concentrations of GLP-1 in rats [43], in agreement with the distribution of GLP-1-secreting cells.

GLP-1 has been associated with an increased activity of peroxisome proliferator-activated receptor (PPAR)-α, and PPARγ [44] and polyphenols have been shown to modulate PPARα expression when added to high-fat diets [45]. PPARα and PPARγ are both known to downregulate the expression of SCD1 and RETSAT [44,46,47]. Our proteomics data show upregulation of SCD1 in the WD compared to ST, which is in accordance with previous studies, showing that SCD1 activity is upregulated in diets high in saturated fatty acids [48]. Interestingly, this upregulation was significantly lower in WD+high-RE compared to WD. High SCD1 activity is associated with hypertriglyceridemia and reduced levels of HDL cholesterol [49]. Consequently, a reduction in the upregulation of the high-fat diet-induced SCD1 expression by RE may contribute to lower concentrations of TG through SCD1. Our data also show RETSAT to be upregulated by WD, but this protein was no longer significantly upregulated when the rats received RE supplements. In humans, hepatic RETSAT expression correlates with serum TGs and steatosis, and depletion of RETSAT in obese mice lowers hepatic and circulating TGs [50]. Consequently, the effect of RE on RETSAT may contribute to the reduction in TG. Several studies have associated lower expression of SCD1 and RETSAT with a favorable lipid profile in plasma and liver tissue, such as decreased plasma triglycerides and increased plasma HDL, lower total liver fat, triglycerides, and cholesterol in the liver, and a higher proportion of unsaturated fatty acids in the liver [47,49,51,52]. The levels of MIF in the liver were increased by RE. MIF has a hepatoprotective role in in vivo models of NAFLD [53]. Hence, it is possible that RE contributed to lower hepatic fat accumulation through its upregulation of MIF.

The observed increase in fasting GLP-1 concentrations might cause an increase in fatty acid β-oxidation and a decrease in fatty acid synthesis, resulting in the reduction in relative fat mass in the liver. Animal studies have reported a decrease in hepatic fat accumulation upon treatment with the GLP-1 receptor agonist exenatide [54,55]. It has also been concluded that rats on a high-fat diet treated with exenatide have increased levels of transcripts of factors involved in fatty acid β-oxidation, resulting in a reduction in total hepatic lipid accumulation [56]. Others have found an increased plasma GLP-1 concentration to be associated with a reduction in mRNA and protein expression of transcription factors required for cholesterol biosynthesis and enzymes that catalyze fatty acid synthesis [57]. Repeating our study protocol in animals deficient in GLP-1 signaling will be required to ultimately test this hypothesis.

Our data showed that rats consuming a high-fat diet for six weeks increased hepatic fat percentage by 2.4-fold, but RE inhibited the accumulation of fat in the liver, resulting in a fat percentage only 1.29-fold higher than ST for WD+high-RE. This finding agrees with that of others who have found polyphenol-rich extracts capable of attenuating steatosis in human and animal models [58,59]. The observations on fat in liver tissue are consistent with our NMR data, where the lipid profile of the liver showed that consumption of a Western-style diet for six weeks resulted in higher levels of cholesterol and triglycerides, and that the present fatty acids were more often saturated fatty acids than in a standard diet. Interestingly, by supplementing the diet with high-RE, the overall lipid profile in the liver shifted to resemble a mix of standard and Western-style diets, with less free and total cholesterol, less triglycerides, and more unsaturated fatty acids than the Western-style diet. This is consistent with previous studies in mice showing that RE can reduce triglyceride levels in the liver by 39% [38]. The rats fed WD supplemented with RE had a relatively increased liver weight. This is most probably associated with the enzyme induction seen due to the xenobiotic metabolism taking place in the liver [60]. This is supported by the increase in several proteins involved in metabolic detoxification pathways, as well as the observed glucuronidation of RE-derived metabolites. 

In our study, we did not see any differences in total body fat mass between ST and the other treatments, even though they were based on a high-fat diet. Interestingly, our WD+high-RE treated rats showed decreased levels of plasma triglycerides and increased levels of HDL, demonstrating a positive effect of continuous consumption of rosemary extract. This contradicts the study by Harach et el. [38], where a RE concentration corresponding to half of the concentration in our WD+high-RE did not lower the plasma triglyceride concentration, although hepatic triglyceride concentration was reduced. We observed higher total cholesterol in plasma in rats fed a diet based on high fat but did not see an effect of RE. A meta-analysis that included 750 participants evaluated the effect of polyphenol-rich cinnamon on blood lipid concentrations and concluded that supplementation with this polyphenol source reduced plasma triglycerides and total cholesterol but did not have any effect on LDL and HDL [61]. In another study, 25 healthy participants received a polyphenol-rich pine bark extract for six weeks, resulting in decreased LDL and increased HDL [62].

Our metabolomics data showed that the concentration of numerous putative BA metabolites in cecal content was inversely proportional to the RE concentration. In contrast to the present study, several studies have shown that polyphenols induce increases in BA excretion [63]. In the present study, the excretion of cholic acid, a primary BA, was increased with increasing addition of RE to the diet. However, the excretion of deoxycholic acid, a secondary BA, and numerous putative BA metabolites were decreased with increasing RE in the diet. Dietary fat stimulates the secretion of BA into the intestinal lumen which promotes changes in the microbiota that increases the conversion of primary to secondary BA [64]. In the present study, this effect was counteracted by the addition of RE which is in agreement with previous studies [65,66], and it has been suggested that polyphenols may regulate the profile and concentration of BA [65] through modulation of the profile of the microflora [67].

Changing the diet from ST to WD and supplementing the high-fat diet with RE caused noticeable changes in the pool of carboxylic acids found in the cecal content, indicating that the microbiota composition was modulated. The considerably lower microbial activity observed when feeding WD is due to the lower content of substrate (dietary fiber) for the microflora in WD. However, the pool of succinic acid increased approximately 10-fold when adding RE to WD and this is in accordance with a previous study showing that the addition of some polyphenols to a high-fat diet increased the concentration of succinic acid in cecal content [68]. It has been shown that RE increases *Prevotella* groups [69], and *Prevotella copri* has been recognized as a succinate producer [70]. Succinate contributes to improved plasma glucose through the activation of intestinal gluconeogenesis [71] and increased glycogen storage in the liver [72], as also seen in our study.

## 5. Conclusions

We did not observe an increase in body weight or total body fat percentage in young, healthy rats during the experimental period when the animals were fed a high-fat diet. However, the treatment with the high-fat diet-induced dyslipidemia manifested as a higher hepatic fat percentage, higher cholesterol and triglyceride concentrations in the plasma and liver, and higher LDL in plasma which was ameliorated when supplementing 4 g RE per kg feed to the diet. Furthermore, RE increased plasma HDL. Our proteomics data suggest this is accomplished through the upregulation of proteins involved in fatty acid metabolism. Altogether, our study strongly indicates that RE improves the lipid profile in animals before the onset of classical signs of obesity, perhaps associated with an increase in fasting GLP-1 or possibly through the upregulation of PPARα and PPARγ. The study indicates that the effect of RE is exerted both via bioavailable metabolites that exert their effect systemically and through gut microbiota-associated metabotypes.

## Figures and Tables

**Figure 1 metabolites-13-00974-f001:**
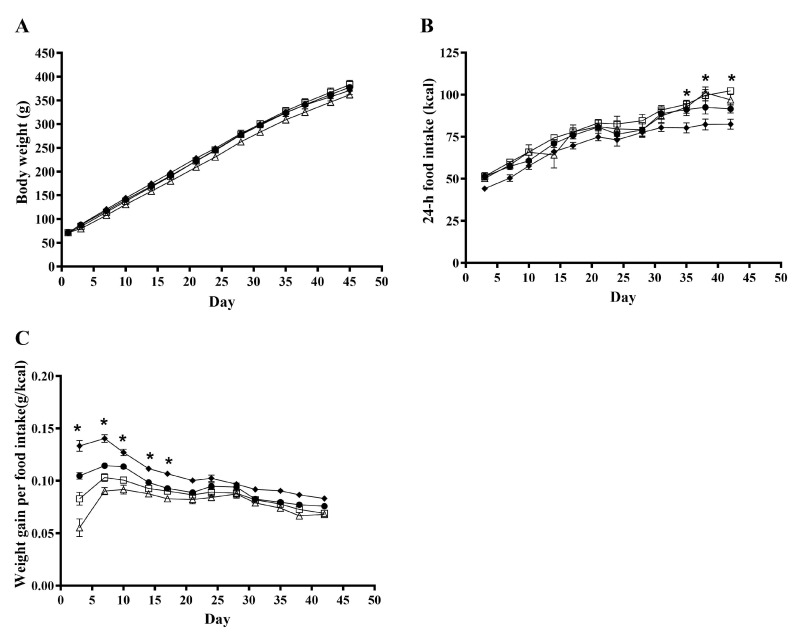
Effects of rosemary extract on body weight, *n* = 16 (**A**), 24-h feed intake, *n* = 8 (**B**), and weight gain per energy intake, *n* = 8 (**C**), in rats fed a ST (♦), WD (●), WD+low-RE (□), or WD+high-RE (∆) over 46 days. Results are presented as mean ± SEM. * indicates a significant difference (*p* < 0.05) between two or more treatments: (**B**) ST and WD+low-RE differ significantly at days 35, 38, and 42, and ST and WD+high-RE differ significantly at day 38. (**C**) All four treatments differ at day 3. In addition, WD and WD+high-RE differ at day 7, and ST and WD+high-RE differ at days 7, 10, 14, and 17. ST, standard diet; WD, Western-style diet; WD+low-RE, WD supplemented with low concentration of rosemary extract; WD+high-RE, WD supplemented with high concentration of rosemary extract.

**Figure 2 metabolites-13-00974-f002:**
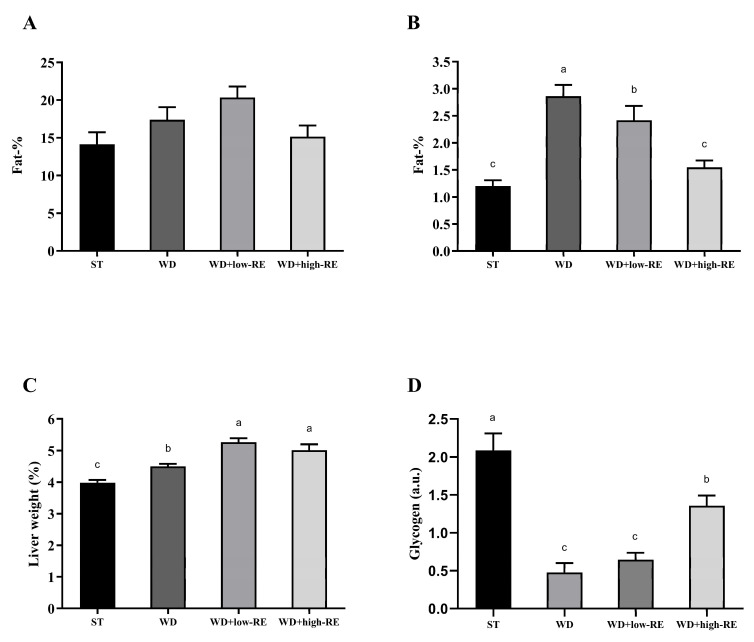
Effects of rosemary extract on (**A**) total body fat percentage, (**B**) liver fat percentage, (**C**) liver weight as a percentage of total body weight, and (**D**) hepatic glycogen (a.u) in rats fed a standard diet (ST), Western-style diet (WD), or a WD supplemented with either low (WD+low-RE) or high (WD+high-RE) concentrations of rosemary extract, measured at day 46. *n* = 8. Results are presented as mean ± SEM. Means without common letters differ significantly (*p* < 0.05).

**Figure 3 metabolites-13-00974-f003:**
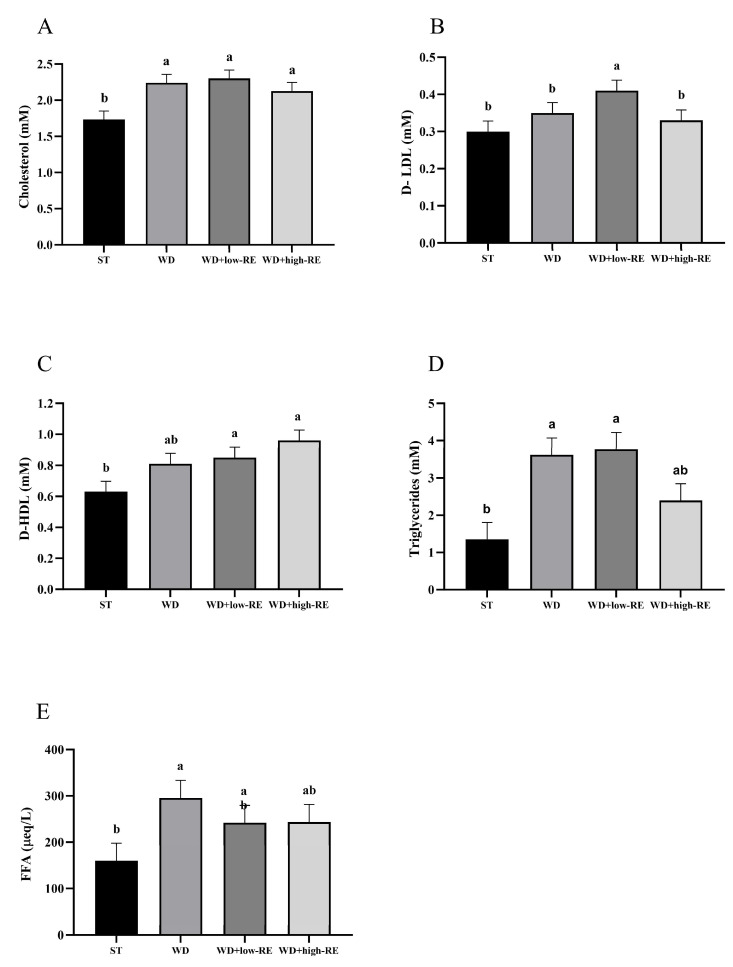
Effects of rosemary extract on plasma levels of (**A**) total cholesterol (mM), (**B**) low-density lipoprotein (D-LDL) (mM), (**C**) high-density lipoprotein (D-HDL) (mM), (**D**) triglycerides (mM), and (**E**) free fatty acids (FFA) (μeq/L) in rats fed a standard diet (ST), Western-style diet (WD), or a WD supplemented with either low (WD+low-RE) or high (WD+high-RE) concentrations of rosemary extract. Measured at day 46. *n* = 8. Results are presented as mean ± SEM. Means without common letters differ significantly (*p* < 0.05).

**Figure 4 metabolites-13-00974-f004:**
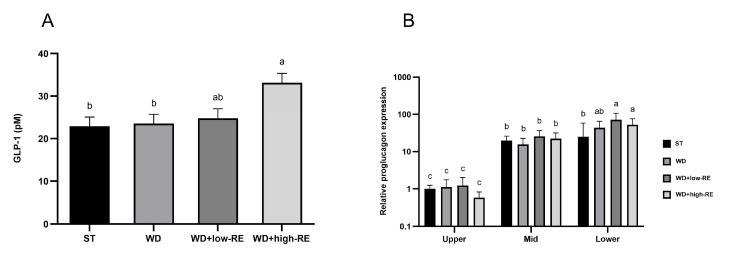
Effects of rosemary extract on (**A**) fasting levels of glucagon-like peptide-1 (GLP-1) (pM) and (**B**) relative proglucagon expression for the interaction between segment and treatment, normalized to ST treatment in upper small intestine (SI) in rats fed a standard diet (ST), Western-style diet (WD), or a WD supplemented with either low (WD+low-RE) or high (WD+high-RE) concentrations of rosemary extract. Measured at day 46. *n* = 8. Results on GLP-1 are presented as mean ± SEM and results on relative proglucagon expression are presented as means with 95% confidence intervals. Means without common letters differ significantly (*p* < 0.05).

**Figure 5 metabolites-13-00974-f005:**
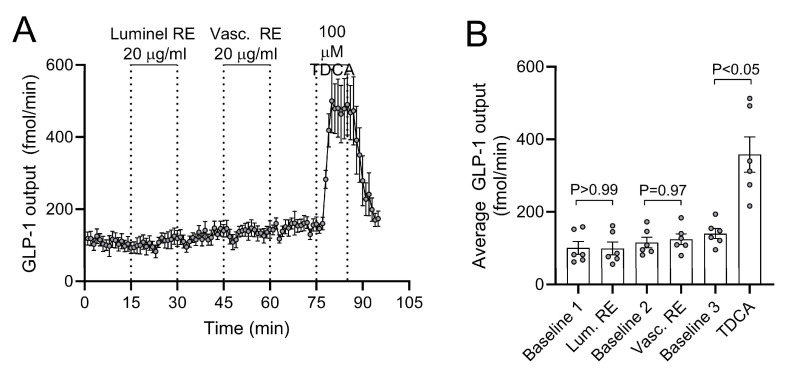
Luminal and vascular rosemary extract (RE)-stimulated glucagon-like peptide-1 (GLP-1) secretion. (**A**) GLP-1 output and (**B**) average GLP-1 output are shown as means ± SEM in response to two subsequent stimulations with 20 µg/mL RE. Intra-luminal infusion of 100µM taurodeoxycholice acid (TDCA) was included at the end of all experiments as a positive control.

**Figure 6 metabolites-13-00974-f006:**
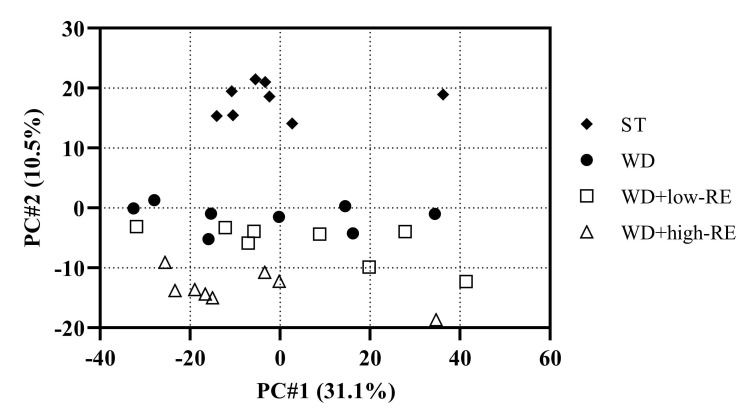
Principal components analysis score plot of liver proteins of rats fed ST (♦), WD (●), WD+low-RE (□), or WD+high-RE (∆), over 46 days. ST, standard diet; WD, Western-style diet; WD+low-RE, WD supplemented with a low concentration of rosemary extract; WD+high-RE, WD supplemented with a high concentration of rosemary extract.

**Figure 7 metabolites-13-00974-f007:**
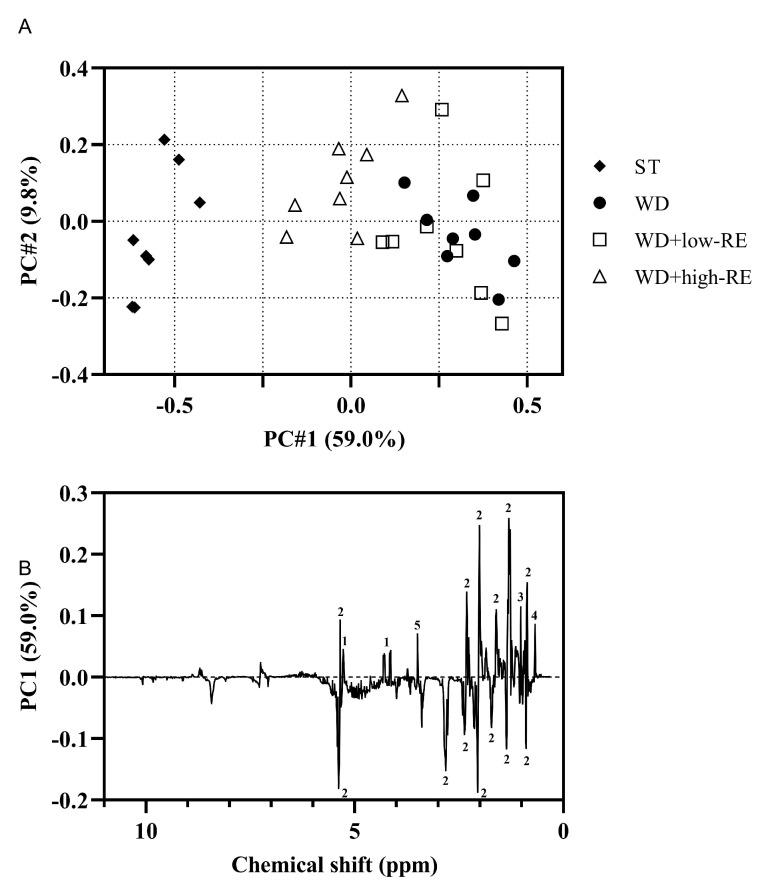
(**A**): Principal components analysis score plot of the non-polar extracts from the liver of rats fed ST (♦), WD (●), WD+low-RE (□), or WD+high-RE (∆), over 46 days. ST, standard diet; WD, Western-style diet; WD+low-RE, WD supplemented with a low concentration of rosemary extract; WD+high-RE, WD supplemented with a high concentration of rosemary extract. (**B**): PC1 loadings plot of non-polar extracts. Assignments: 1, glycerol backbone; 2, fatty acyl chain; 3, free cholesterol; 4, total cholesterol; 5, residual methanol.

**Figure 8 metabolites-13-00974-f008:**
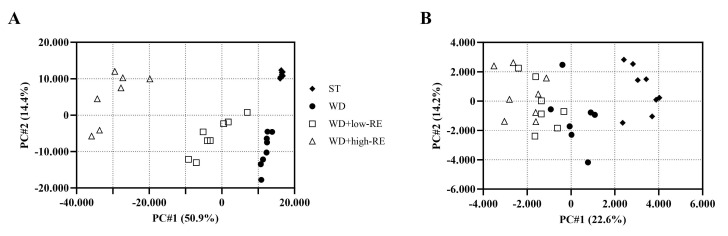
Principal components analysis score plot of the metabolome, acquired in negative mode, of cecal content (**A**) and plasma (**B**) of rats fed ST (♦), WD (●), WD+low-RE (□), or WD+high-RE (∆), over 46 days. ST, standard diet; WD, Western-style diet; WD+low-RE, WD supplemented with a low concentration of rosemary extract; WD+high-RE, WD supplemented with a high concentration of rosemary extract.

**Table 1 metabolites-13-00974-t001:** Gene and protein name of quantified proteins involved in fatty acid metabolism. Fold changes between the different diets in relation to rats fed a standard diet (ST), Western-style diet (WD), or a WD supplemented with either low (low-RE) or high (high-RE) concentrations of rosemary extract, measured at day 46. *n* = 8.

Protein Name	Gene	WD/ST	WD+Low-RE/ ST	WD+High-RE/ ST
Arylacetamide deacetylase	*AADAC*	1.7	1.6	1.7
ATP-binding cassette sub-family D member 3	*ABCD3*	1.2	1.3	1.5
Short/branched chain specific acyl-CoA dehydrogenase, mitochondrial	*ACADS*	0.5	0.6	0.5
Aldehyde oxidase 1	*AOX1*	1.2	1.2	1.3
CDGSH iron-sulfur domain-containing protein 1	*CISD1*	0.9	0.8	0.8
2,4-dienoyl-CoA reductase, mitochondrial	*DECR1*	2.0	1.6	1.8
Enoyl-CoA delta isomerase 1, mitochondrial	*ECI1*	2.1	1.8	2.2
Hydroxyacyl-coenzyme A dehydrogenase, mitochondrial	*HADH*	1.3	1.0	1.0
Trifunctional enzyme subunit alpha, mitochondrial	*HADHA*	1.4	1.3	1.5
Trifunctional enzyme subunit beta, mitochondrial	*HADHB*	1.3	1.0	1.3
Hydroxymethylglutaryl-CoA synthase, mitochondrial	*HMGCS2*	1.4	1.2	1.2
Corticosteroid 11-beta-dehydrogenase isozyme 1	*HSD11B1*	0.5	0.3	0.5
Estradiol 17-beta-dehydrogenase 2	*HSD17B2*	1.2	1.4	1.6
MICOS complex subunit Mic60 (Fragment)	*IMMT*	0.8	1.2	1.7
Enoyl-[acyl-carrier-protein] reductase, mitochondrial	*MECR*	1.6	2.2	4.1
Macrophage migration inhibitory factor	*MIF*	1.1	1.3	1.6
5′-AMP-activated protein kinase catalytic subunit alpha-2	*PRKAA2*	1.4	1.2	1.5
All-trans-retinol 13,14-reductase	*RETSAT*	2.4	1.4	1.2
Stearoyl-coenzyme A desaturase 1	*SCD1*	9.8	10.1	7.6

**Table 2 metabolites-13-00974-t002:** Weight of cecal content and pool size of total and individual carboxylic acids in caecum content in rats fed a standard diet (ST), Western-style diet (WD), or a WD supplemented with either low (low-RE) or high (high-RE) concentrations of rosemary extract, measured at day 46. *n* = 8. Mean values with 95% confidence intervals.

	ST	WD	WD+Low-RE	WD+High-RE	*p*-Value
Cecal content (g)	4.64 ^a^ [3.69–5.83]	1.79 ^c^ [1.42–2.25]	2.44 ^b^ [1.94–3.07]	2.62 ^b^ [2.01–3.41]	<0.0001
Carboxylic acids, pool size (µmol)					
Total carboxylic acids	555 ^a^ [445–692]	84 ^c^ [67–104]	125 ^b^ [100–155]	128 ^b^ [96–169]	<0.0001
Acetic acid	302 ^a^ [241–379]	52.3 ^b^ [41.7–65.5]	69.0 ^b^ [55.1–86.5]	69.5 ^b^ [52.3–92.5]	<0.0001
Propionic acid	58.9 ^a^ [44.7–77.5]	13.2 ^b^ [10.0–17.3]	15.5 ^b^ [11.8–20.4]	12.7 ^b^ [9.0–18.0]	<0.0001
Butyric acid	165 ^a^ [128–214]	6.75 ^b^ [5.21–8.74]	3.15 ^c^ [2.43–4.08]	1.72 ^d^ [1.27–2.32]	<0.0001
Succinic acid	4.03 ^b^ [2.09–7.75]	2.45 ^b^ [1.41–4.26]	31.6 ^a^ [18.9–53.1]	21.8 ^a^ [12.0–39.7]	<0.0001

^a, b, c, d^ Mean values within a row with unlike superscript letter were significantly different (*p* < 0.05).

## Data Availability

None of the data were deposited in an official repository. The data that support the study findings are available from the authors upon request.

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
