# Peer review of "Unravelling Effects of Rosemary (Rosmarinus officinalis L.) Extract on Hepatic Fat Accumulation and Plasma Lipid Profile in Rats Fed a High-Fat Western-Style Diet"

_metabolites, 2023, doi:10.3390/metabo13090974_

Round 1

Reviewer 1 Report

Unravelling effects of rosemary (Rosmarinus officinalis L.) ex-2 tract on hepatic fat accumulation and plasma lipid profile in 3 rats fed a high-fat Western style diet

The authors have reported in this paper the effects of Rosmarinus officinalis on clinical and metabolic changes in rats that have been fed a high fat Western style diet in order to evaluate its effects. There is significant interest in this study and it provides some information about the R. officinalis biological potentials.

In abstract, the treatments did not affect body weight. According to this study, both WD and ST diets resulted in similar body weight increases throughout the course of the experiment. In the figure at the bottom, we can see that the body weights of the experimental animals were similar to those of the control animals. There is some controversy about this because High fat diets can cause an increase in body weight compared to the ST diets, it is a fact. Could you please clarify the statements you have made.

 Furthermore, high levels of succinic acid in the cecal content of RE treated animals suggested a modulation 29 of the microbiota composition. As of now, I have not found any data that is related to the dynamics of microbial communities in experimental animals. In that case, how do authors claim that microbiota composition is modulated?

We hypothesized that RE reduces feed intake in rats and inhibits diet-induced weight gain through increased fasting GLP-1 levels. Please find the data presented in figure 1A for your reference. There is no difference in body weight between experimental animals in the different periods of the experiment.

There is a lack of clarity regarding the design and presentation of animal experiments. It would be helpful if the experimental groups and treatment methods could be clearly defined.

The experiment design and the protocol in general need to be improved by several ways, including reducing the length of the write up part or providing references if the protocol is a common one.

ST treatment resulted in the lowest relative liver weight, it increased when feeding WD and adding RE to the diet increased the relative liver weight further.

What does it mean when there is an increase in liver weight similar to that of treatment for WD?

Increases in liver weight are associated with an increase in fat deposition in the liver.

 Would it be possible for you to explain the reason for the increase in liver weight?

For mean plasma levels of LDL-cholesterol (Figure 3B), only WD+low-363 RE treatment resulted in a significantly higher level than ST

Is it true that low doses of the RE could have the ability to increase the levels of LDL in the blood?

Analysis of mean plasma levels of HDL-choles terol (Figure 3C) showed that WD+low-RE and WD+high-RE treatment caused significant 366 higher plasma HDL levels than ST treatment. Nevertheless, there is no significant difference in the outcome of HFD treatment as compared to RE treatment. In this case, what is the study's purpose?

Is it possible for the author to explain the differences between the GLP output of RE treated and non-treated GLP?

There are no significant differences between baseline stimulation and RE stimulation of GLP levels.

The table 2 is not clear enough; it should be made clearer and easier to understand.

A significant improvement is required in the quality of data presented in the result section as well as its interpretation, along with the discussion section providing essential information.

Unravelling effects of rosemary (Rosmarinus officinalis L.) ex-2 tract on hepatic fat accumulation and plasma lipid profile in 3 rats fed a high-fat Western style diet

The authors have reported in this paper the effects of Rosmarinus officinalis on clinical and metabolic changes in rats that have been fed a high fat Western style diet in order to evaluate its effects. There is significant interest in this study and it provides some information about the R. officinalis biological potentials.

In abstract, the treatments did not affect body weight. According to this study, both WD and ST diets resulted in similar body weight increases throughout the course of the experiment. In the figure at the bottom, we can see that the body weights of the experimental animals were similar to those of the control animals. There is some controversy about this because High fat diets can cause an increase in body weight compared to the ST diets, it is a fact. Could you please clarify the statements you have made.

 Furthermore, high levels of succinic acid in the cecal content of RE treated animals suggested a modulation 29 of the microbiota composition. As of now, I have not found any data that is related to the dynamics of microbial communities in experimental animals. In that case, how do authors claim that microbiota composition is modulated?

We hypothesized that RE reduces feed intake in rats and inhibits diet-induced weight gain through increased fasting GLP-1 levels. Please find the data presented in figure 1A for your reference. There is no difference in body weight between experimental animals in the different periods of the experiment.

There is a lack of clarity regarding the design and presentation of animal experiments. It would be helpful if the experimental groups and treatment methods could be clearly defined.

The experiment design and the protocol in general need to be improved by several ways, including reducing the length of the write up part or providing references if the protocol is a common one.

ST treatment resulted in the lowest relative liver weight, it increased when feeding WD and adding RE to the diet increased the relative liver weight further.

What does it mean when there is an increase in liver weight similar to that of treatment for WD?

Increases in liver weight are associated with an increase in fat deposition in the liver.

 Would it be possible for you to explain the reason for the increase in liver weight?

For mean plasma levels of LDL-cholesterol (Figure 3B), only WD+low-363 RE treatment resulted in a significantly higher level than ST

Is it true that low doses of the RE could have the ability to increase the levels of LDL in the blood?

Analysis of mean plasma levels of HDL-choles terol (Figure 3C) showed that WD+low-RE and WD+high-RE treatment caused significant 366 higher plasma HDL levels than ST treatment. Nevertheless, there is no significant difference in the outcome of HFD treatment as compared to RE treatment. In this case, what is the study's purpose?

Is it possible for the author to explain the differences between the GLP output of RE treated and non-treated GLP?

There are no significant differences between baseline stimulation and RE stimulation of GLP levels.

The table 2 is not clear enough; it should be made clearer and easier to understand.

A significant improvement is required in the quality of data presented in the result section as well as its interpretation, along with the discussion section providing essential information.

Author Response

Dear reviewer,

Thank you for your careful review of our manuscript. Please find our responses to the points raised in the attachment.

Best regards,

Mette Skou Hedemann

Reviewer 2 Report

 Minor editing of English language required

Author Response

Dear reviewer,

Thank you for the thorough review of our manuscript. Please find our responses to the points raised by you in the attachment.

Best regards,

Mette Skou Hedemann
